# INTERNAL-CONSISTENCY CONSTRAINTS FOR EMERGENT COMMUNICATION

## ABSTRACT

When communicating, humans rely on internally-consistent language representations. That is, as speakers, we expect listeners to behave the same way we do when we listen. This work proposes several methods for encouraging such internal-consistency in dialog agents in an emergent communication setting: shared embeddings, symmetric encoding/decoding, and self-play. We consider two hypotheses about the effect of internal-consistency constraints: 1) that they improve agents' ability to refer to unseen referents, and 2) that they improve agents' ability to generalize across communicative roles (e.g. performing as a speaker despite only being trained as a listener). While we do not find evidence in favor of the former, our results show significant support for the latter. In particular, we show that, when self-consistency is enforced via self-play, agents are able to perform in novel roles as well as if they were trained with direct supervision in those roles.

## 1 INTRODUCTION

Emergent communication is the study of how linguistic protocols evolve when agents are tasked to cooperate. For example, agents engaged in a simple object retrieval task learn to communicate with one another in order to get the items they want (Lazaridou et al., 2018). To date, work of this type has each agent assume a conversational role. Thus, agents are often trained only to speak or only to listen (Lazaridou et al., 2018), or similarly trained to speak using a vocabulary disjoint from the vocabulary it is understands as a listener–e.g. speaking only to ask questions (*"what color?"*) and listening only to comprehend the answer (*"blue"*) (Kottur et al., 2017; Das et al., 2017).

These assumptions are misaligned with how we think about human communication, and with the way we'd like computational models to work in practice. As humans, not only can we easily shift between roles, we also know that there is inherent symmetry between these roles: we expect others to speak (or listen) similarly to the way we do, and we know that others expect the same of us.

We test if dialog agents that incorporate the symmetry between themselves and their communicative partners learn more generalizable representations than those which do not. We introduce three modifications to the agents to encourage that they abide by the "golden rule": speak/listen as you would want to be spoken/listened to. Specifically, these modifications include self-play training objectives, shared embedding spaces, and symmetric decoding and encoding mechanisms that share parameters. We test two hypotheses about the effect of the proposed modifications on emergent communication:

1. Internal-consistency constraints improve agents' ability to generalize to unseen items–e.g. training on *"red square"* and *"blue circle"* and then testing on *"blue square"*.

2. Internal-consistency constraints improve agents' ability to generalize across communicative roles–e.g. training on *"blue"* as a listener, and using *"blue"* as a speaker when testing.

We evaluate the effect of each of the proposed modifications with two reference game datasets and two model architectures, an RNN model used by Lazaridou et al. (2018) and a Transformer model. We find no evidence to support that internal-consistency improves generalization to unseen items (Hypothesis 1), but significant evidence that these proposed constraints enable models to generalize learned representations across communicative roles (Hypothesis 2), even in the case of where the agent receives no direct training in the target (test) role. All of our code and data are available at `bit.ly/internal-consistency-emergent-communication`.

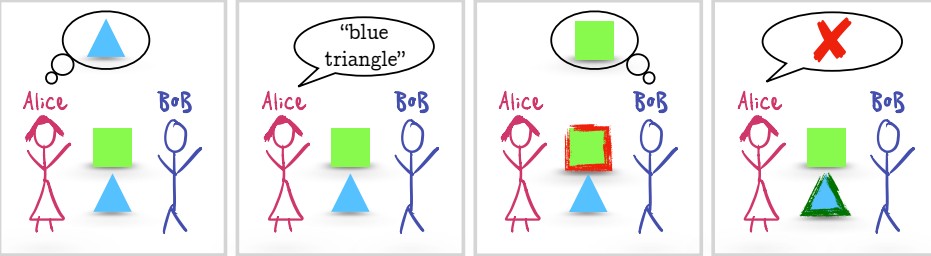

Figure 1: **Reference game.** Alice wants the blue triangle, and asks Bob to get it for her. He selects the green square, and Alice tells him "No" and shows him the item she wanted.

## 2 EMERGENT COMMUNICATION VIA REFERENCE GAMES

Following past work on emergent communication (Lazaridou et al., 2018), we evaluate our models in a cooperative *reference game* (depicted in Fig. 1)(Lewis, 2008). A basic reference game consists of two agents and a set of items (each a vector of `attribute:value` pairs). One agent (the "speaker") has a target item in mind, and must describe it to the other agent (the "listener"). After training, we are primarily interested in the properties and the generalizability of the resulting *lexicon*: the mapping from symbols in the vocabulary onto items and/or their attributes.

Implemented naively, agents converge to a trivial lexicon which assigns a unique symbolic name to each item in the training set (Kottur et al., 2017). Thus, a challenge in emergent communication is to build agents which learn a lexicon that maps symbols to attributes or other primitive concepts, so the agents can generalize to novel combinations of attributes at test time. For example, if trained on items {[`shape:circle, color:red`], [`shape:square, color:blue`]}, an ideal lexicon would map each attribute-value pair to a unique word (e.g. *word1*=`blue`, *word2*=`circle`) so the agents could refer to an unseen combination ([`shape:circle, color:blue`]) at test time.

Like past work, this paper focuses on evaluating inductive biases that result in lexicons which generalize to novel items. In addition, we focus on a different type of generalization: generalization across communicative roles. Prior work has tended to treat these roles as distinct, such that an agent may learn to comprehend *"blue"* and *"circle"* perfectly as a listener, and yet be unable to refer to [`shape:circle, color:blue`] as a speaker. Thus, we test whether models that are trained in both roles (either with direct supervision, or only via self-play) learn lexicons that not only generalize to novel items but also which can be transferred across roles.

**Notation.** The space of possible references is parameterized by the number of attributes $n_f$ that describe each item (e.g. `color`) and the number of values $n_v$ each attribute can take (e.g.{`red, blue`}). Each item $o$ is a bag-of-features vector $o \in \{0,1\}^N$ where $N = n_f \cdot n_v$. Each index $o_i$ is 1 if $o$ expresses the $i$th feature value. The speaker produces a message with symbols from a vocabulary $\mathcal{V}$ with length $L$. For comparison, we use the best-performing setting $|\mathcal{V}| = 100$ and $L = 10$ from previous work (Lazaridou et al., 2018). Symbols in $\mathcal{V}$ are represented as 1-hot vectors. In each round of the reference game, we construct $\langle C, r, r \rangle$ where $C$ is the context (set of item column vectors stacked into a matrix), $r$ is a vector representing the referent, and $r$ is the index of the referent in $C$. We uniformly sample $k-1$ items as distractors to form $C = \{o_1, \ldots o_{k-1}\} \cup \{r\}$. The distractors are is sampled randomly each round (in every epoch).

## 3 MODELS

We begin with a general architecture and training objective to underly all of our models (Sections 3.1 and 3.2). We then introduce three modifications which can be used to encourage internally-consistent representations: a self-play training objective, a shared embedding space, and a symmetric decoding and encoding mechanism with shared parameters (Section 3.3) [1].

---

[1]For implementation details, refer to Appendices A.1, A.2, A.3, or source code `bit.ly/ internal-consistency-emergent-communication`.

### 3.1 ARCHITECTURE

Agents contain four modules. **Embedding modules** 1) $E_{item} \in \mathbb{R}^{N \times D}, E_{message} \in \mathbb{R}^{|\mathcal{V}| \times D}$. $E_*(x)$ embed items and messages. When speaking, the **decoder module** 2) consumes the embedded referent $E_{item}(\boldsymbol{r})$ and produces a discrete message $\boldsymbol{M} \in \mathcal{V}^L$. Next, when listening, the **encoder module** 3) consumes embedded messages $E_{message}(\boldsymbol{M}) \in \mathbb{R}^{W \times D}$ and then produces a representation of the referent $\hat{\boldsymbol{r}} \in \mathbb{R}^D$. Finally, a non-parametric **pointing module** 4) produces a distribution $P(\boldsymbol{C})$ over the context by matrix multiplying $\hat{\boldsymbol{r}}$ with the embedded context $E_{item}(\boldsymbol{C})$.

The decoders emit one symbol at a time, auto-regressively producing fixed-length messages. The messages are discretized with the straight-through Gumbel Softmax (Jang et al., 2016) as in Mordatch & Abbeel (2018). This converts a distribution to a one-hot vector while permitting gradients to flow through the operation and enables the agents to be optimized without reinforcement methods.

The **Recurrent model** uses a LSTM (Hochreiter & Schmidhuber, 1997) decoder when speaking and a LSTM encoder when listening, as in (Lazaridou et al., 2018). The **Transformer model** (Vaswani et al., 2017) uses a Transformer Decoder when speaking and a Transformer Encoder to encode when listening. See Appendix A for implementation details and hyperparameters.

### 3.2 TRAINING OBJECTIVES

Speaking $\mathscr{S} : \boldsymbol{r}, \theta \rightarrow \boldsymbol{M}$ and listening $\mathscr{L} : \boldsymbol{C}, \boldsymbol{M}, \theta \rightarrow P(\boldsymbol{C})$ are both functions where $\boldsymbol{M} \in \mathcal{V}^L \subset \{0, 1\}^{L \times |\mathcal{V}|}$ is a discrete-valued message with length $L$, $P(\boldsymbol{C})$ is a distribution over the items in context, and $\theta$ are optimizable parameters. We optimize the parameters $\theta_A, \theta_B$ of agents $A, B$ over a dataset $D$ to select each referent $r$ from among the distractors in its context $\boldsymbol{C}$ by minimizing the negative log likelihood of selecting the correct referent in context (Eq 1). In our experiments, the speaker modules and the listener modules instantiate the function $\mathscr{S}$ and $\mathscr{L}$ respectively.

$$\mathcal{L}_{A \rightarrow B} = \sum_{\langle \boldsymbol{C}, \boldsymbol{r}, r \rangle \in D} -\log \mathscr{L}(\boldsymbol{C}, \mathscr{S}(\boldsymbol{r}; \theta_A); \theta_B)_r, \tag{1}$$

### 3.3 IMPOSING INTERNAL-CONSISTENCY CONSTRAINTS

We investigate three internal-consistency constraints, that encourage internally-consistent representations. Baseline agents consist of two separate sets of parameters, one for listening and one for speaking. For example, the baseline recurrent model consists of two recurrent models. This corresponds to the scenario where agents either speak or listen, but not both (Lazaridou et al., 2018).

We introduce a 1) **self-play loss** for both agents of the same form as Eq. 1, except the given agent fulfills both roles, encouraging it to speak/listen to others the same way it speaks/listens to itself. When we use the self-play training objective, we use it for both agents. Next, 2) **shared embedding** agents use the same item embeddings and the same message embeddings when listening and speaking. Finally, 3) **symmetric** encoding and decoding agents use the same parameters (but different mechanisms) to decode a message when speaking as it does to encode a message when listening. Parameters are only ever shared between roles *within an agent*, and never between agents.

## 4 DATASETS

Our evaluation is based on the simple reference game[2] as described in Section 2, played across two datasets. The datasets, summarized in Table 1, target different aspects of lexical reference. The first, Visual Attributes for Concepts (CONCEPTS) Silberer et al. (2013), is derived from annotated images of actual items (animals, tools, etc). Thus, the data contains realistic co-occurance patterns: groups of attributes like `has-head`, `has-mouth`, and `has-snout` appears together several times, whereas `has-seeds`, `has-mouth`, `made-of-metal` never co-occur. The intuition is that a good lexicon will exploit the structure by developing words which refer to groups of frequently co-occurring attributes (e.g. *"mammal"*) and will describe unseen referents in terms of

---

[2]We use a "context-hidden" game: the context is visible to the listener but not the speaker. This is to ensure that the speaker does not have access to additional attributes (e.g. position of item within the context) that could enable the agents to find degenerate solutions to the game.

these primitive concepts. The second dataset, SHAPES, is one we create ourselves in contrast with the CONCEPTS data. In SHAPES, items correspond to abstract shapes which are exactly describable by their attributes (e.g. `blue`, `shaded`, `hexagon`). All the attributes are independent and there is no co-occurence structure to exploit, so a good lexicon should ideally provide a means for uniquely specifying each attribute's value independently of the others.

| Name | Features ($n_f$) | Values ($n_v$) | Context Size ($k$) | Train | Val | Test |
|---|---|---|---|---|---|---|
| SHAPES | 10 | 50 | 5 | 1000 | 200 | 100 |
| CONCEPTS | 597 | 1 | 5 | 375 | 50 | 85 |

Table 1: Dataset statistics.

## 5 RESULTS

We present experimental results aimed at testing the hypotheses stated in Section 1. To provide intuition, we frame experiments in terms of two agents, "Alice" (Agent A) and "Bob" (Agent B), who are taking turns asking for and fetching toys.

### 5.1 HYPOTHESIS 1: GENERALIZATION TO NEW ITEMS

#### 5.1.1 SETUP

We first test whether any of the proposed internal-consistency constraints improve the agents' ability to generalize to novel items–i.e. items which consist of unseen combinations of features. Here, we focus on the performance when models are trained and tested *within the same communicative role*. This corresponds to the setting that has typically been used in prior work on emergent communication: Alice always speaks in order to ask for toys, Bob always responds by fetching them, and the pair's success is evaluated in terms of Alice's ability to describe new toys such that Bob correctly gets them. For evaluation, we hold out a subset of the value combinations from each dataset to use for testing. For example, the agents might be trained to refer to {[`shape:circle`, `color:red`],[`shape:square`, `color:blue`]} and then tested on its ability to refer to [`shape:circle`, `color:blue`]. We compute validation and test accuracies.[3]

#### 5.1.2 COMPARISON OF MODEL ARCHITECTURES

Before introducing our internal-consistency constraints, we first evaluate our various architectures in this vanilla setting. Thus, these results serve chiefly to calibrate differences between model architectures. If we see differences between the test conditions, these should be attributed to general architectural advantages associated with implementing these internal-consistency constraints, rather than interpreted as evidence for/against internal-consistency *per se*. Table 2 shows the results. Overall, the transformer architecture outperforms the recurrent architecture, and the transformer outperforms the previously best-reported result for this task (which is only available on the CONCEPTS data). We observe no clear trend associated with the shared embedding module (sometimes it helps, sometimes it hurts), but do see a slight positive effect associated with the symmetric transformer.

#### 5.1.3 EFFECTS OF INTERNAL-CONSISTENCY CONSTRAINTS.

We next look at whether there is an advantage to using the the internal-consistency constraints, even when agents remain in fixed conversational roles. Intuitively, this corresponds to a scenario in which both Alice and Bob are capable of performing either role (speaking or listening), but nonetheless, they only ever interact within the same routine: Alice only every speaks and Bob only ever listens and fetches items in response. However, both Alice and Bob can imagine how they would behave if they were to assume the opposite role, and thus, via self-play, each can enforce internal-consistency between their own actions as speaker (listener) and the way they imagine they would respond as listener (speaker). In this manner, although Alice and Bob only ever receive direct feedback in

---

[3]We additionally tried computing Tree Reconstruction Error as introduced by Andreas (2019). See Appendix A.5 for implementation details. However, we did not find an interesting differences across conditions. Therefore, for compactness, we omit these results.

| | SHAPES | | CONCEPTS | |
| --- | --- | --- | --- | --- |
| | Val | Test | Val | Test |
| Random (Baseline) | 20.0 | 20.0 | 20.0 | 20.0 |
| Prior SOTA (Lazaridou et al., 2018) | - | - | - | 81.6 |
| Recurrent Model | $56.8 \pm 0.8$ | $49.0 \pm 1.9$ | $67.9 \pm 1.6$ | $70.1 \pm 2.6$ |
| Transformer Model | $86.5 \pm 1.5$ | $89.5 \pm 1.2$ | $86.2 \pm 1.7$ | $87.0 \pm 0.7$ |
| Symmetric Transformer Model | $86.7 \pm 1.2$ | $89.3 \pm 1.4$ | $92.7 \pm 0.7$ | $89.3 \pm 1.6$ |

Table 2: **Results** when agents are trained and tested in a single role, before any internal-consistency constraints. These scores are mean accuracy with 95% confidence range averaged over 5 runs over different test set samplings (the distractors change).

one role, they can still impose internally consistent behavior across both roles. It is conceivable that doing so might improve performance even though each agent remains in a fixed role, either by providing the model with additional information about the task structure, or simply by acting as a regularizer. Thus, for completeness, we assess whether the internal-consistency constraints provide any advantage to the models in the vanilla emergent communication setting.

Table 3 shows the effect of adding the self-play objective in the fixed-role setting, across architectures and datasets. The trends are mixed: it appears the additional signal only noises the baseline and symmetric models, whereas the shared embeddings models are able to leverage it effectively. Thus, the effect is not clear enough to establish conclusively that the internal-consistency constraints help the agents generalize in this fixed-role setting, and in fact it may hurt.

| | Baseline | +Self-play | | +Shared Emb. | | +Symmetric | |
| --- | --- | --- | --- | --- | --- | --- | --- |
| | % | % | $\Delta$ | % | $\Delta$ | % | $\Delta$ |
| RNN, Shapes | 49.0 | $52.6 \pm 0.8$ | $+3.6$ | $69.0 \pm 2.4$ | $+20.0$ | $20.1 \pm 1.5$ | $-28.9$ |
| RNN, Concepts | 70.1 | $68.6 \pm 2.2$ | $-1.5$ | $81.4 \pm 1.9$ | $+11.3$ | $20.6 \pm 1.3$ | $-49.5$ |
| Trans, Shapes | 89.5 | $78.7 \pm 1.7$ | $-10.8$ | $86.3 \pm 1.0$ | $-3.2$ | $87.4 \pm 0.4$ | $-2.1$ |
| Trans, Concepts | 87.0 | $85.3 \pm 1.4$ | $-1.7$ | $89.0 \pm 1.3$ | $+2.0$ | $90.1 \pm 0.4$ | $+3.1$ |

Table 3: Performance on task of referring to/fetching unseen items for baseline model compared against models with the internal-consistency constraints. To highlight the difference of each constraint compared to the baseline performance, *each delta compares the performance of the modified model to the baseline model*. In this setting, we see no clear advantage to enforcing internal-consistency via self-play. These scores are mean accuracy with 95% confidence interval averaged over 5 runs over different test set samplings (the distractors change).

## 5.2 HYPOTHESIS 2: GENERALIZATION TO NEW ROLES

### 5.2.1 SETUP

We now look at whether internal-consistency improves the agents' ability to generalize linguistic knowledge across roles. For example, we can picture the following scenario: Alice is speaking to Bob, and asks for the *"truck"*. Bob hands her the doll, and Alice replies negatively, indicating that what she actually wanted was the truck. Now, without additional direct supervision, when Bob wants the truck, will he know do use the word *"truck"*? Such a setting is particularly relevant in practical settings, for example when robotic agents must reach high accuracy despite only limited access to human interaction for training. We consider two versions of this setting, involving different levels of direct supervision (i.e. interaction with the other agent) as described below.

**Training in one role.** Our first experimental setting assumes that Alice and Bob each only receive direct training in one role, e.g. Alice only ever speaks to Bob, so Alice only receives feedback on how she is performing as a speaker, and Bob on how he is performing as a listener. However, both Alice and Bob are able to practice in the opposite role via self-play. This setup is analogous to the experiment just discussed in Section 5.1.3. However, unlike before, Alice and Bob will be tested in

the roles opposite of those in which they were trained. That is, if Alice was trained as a speaker, then she will be tested as a listener (on her ability to correctly identify items to which Bob refers).

**Training in both roles.** In our second experimental setting, we assume Alice and Bob enjoy a healthy friendship, in which both take turns speaking and listening to each other, and thus both receive direct supervision in both roles. However, they do not necessarily receive equal training on every vocabulary item. Rather, there are some contexts in which Alice only speaks and other contexts in which she only listens. Intuitively, this corresponds to a scenario in which Alice speaks exclusively about food (while Bob listens), while Bob speaks exclusively about toys (while Alice listens). We are interested in testing how well Alice is able to speak about toys at test time.

We use the SHAPES dataset[4] to create two training splits, each having the same attributes but covering disjoint sets of values. For example, the first training split (`train-1`) might have color∈{`blue`, `red`, `yellow`} whereas the second training split (`train-2`) has color∈{`green`, `orange`, `purple`}. We use `train-1` to train Alice as speaker and Bob as listener and `train-2` to train them in the reverse roles. We then report performance with Alice as listener and Bob as speaker using a test set that uses the same attribute values as `train-1`.

### 5.2.2 RESULTS

Our results for both training conditions are shown in Table 4. The baseline model (which includes no internal-consistency constraints) performs, unsurprisingly, at chance. Adding the self-play objective gives improvements across the board. Again, while seemingly straight forward, this result has promising practical interpretations in settings in which a model has access to only a small amount of interaction. For example, a human may be willing to train a robot via speaking (pointing and naming items), but not patient enough to train it via listening (responding to the robot's noisy commands). In such a setting, the ability to massively augment performance via self-play is significant. In addition to the self-play objective, we see that enforcing shared-embedding spaces yields further significant performance gains (in the range of 30 percentage points in some cases). The symmetric constraints on top of self-play and shared embeddings seem to hurt performance in general.[5]

| | Baseline | +Self-play | | +Shared Emb. | | +Symmetric | |
|---|---|---|---|---|---|---|---|
| **One Role** | % | % | Δ | % | Δ | % | Δ |
| RNN, Shapes | $19.8 \pm 1.3$ | $50.7 \pm 1.3$ | $+30.7$ | $70.7 \pm 1.8$ | $+50.9$ | $20.1 \pm 1.9$ | $-49.9$ |
| RNN, Concepts | $21.3 \pm 2.3$ | $62.9 \pm 0.9$ | $+42.9$ | $81.7 \pm 0.8$ | $+18.8$ | $19.6 \pm 1.5$ | $-62.1$ |
| Trans, Shapes | $19.5 \pm 0.8$ | $82.1 \pm 1.5$ | $+62.6$ | $79.5 \pm 0.8$ | $-2.6$ | $83.9 \pm 1.8$ | $+4.4$ |
| Trans, Concepts | $19.6 \pm 2.9$ | $52.8 \pm 1.7$ | $+32.6$ | $80.3 \pm 1.1$ | $+27.5$ | $77.3 \pm 2.1$ | $-3.0$ |
| **Both Roles** | % | % | Δ | % | Δ | % | Δ |
| RNN, Shapes | $20.4 \pm 1.0$ | $40.1 \pm 0.6$ | $+19.7$ | $18.8 \pm 1.5$ | $-21.3$ | $69.4 \pm 1.9$ | $+50.6$ |
| Trans, Shapes | $20.9 \pm 2.5$ | $61.7 \pm 2.2$ | $+40.8$ | $70.3 \pm 2.0$ | $+8.6$ | $74.7 \pm 1.5$ | $+4.4$ |

Table 4: Performance for tasks that requires agents to generalize across roles–e.g. training on the word *"blue"* as a listener, but then having to produce *"blue"* as a speaker. "One Role" refers to when agents receive direct feedback in a single role (i.e. their training on the other roles is only via self-play). "Both Roles" refers to when agents receive direct feedback in both roles, but only see the test vocabulary in the role opposite that in which they are tested. To inspect the additive differences between the internal-consistency constraints, each delta compares the performance of the current column to the previous column. These scores are mean accuracy with 95% confidence range averaged over 5 with different test set samplings (the distractors change).

### 5.3 TAKEAWAYS

Overall, when agents can be trained directly in the role in which they are tested, there is no clear evidence that adding internal-consistency constraints improves the ability of agents to generalize

---

[4]We cannot construct analogous splits with CONCEPTS since the train set is small.

[5]We found that, in absence of self-play, neither symmetric encoding/decoding nor shared embeddings performed competitively. Therefore, for compactness, we only report the results of the constraints in combination.

to new items. However, internal-consistency constraints improve performance significantly when agents have limited ability to train in a given role. Specifically, models which are equipped with self-play training objectives and shared embedding spaces show superior ability to generalize learned representations across roles, and perform about as well as if they had been trained on the target role.

## 6 ANALYSIS

In this section we provide additional analyses to highlight the effects of internal-consistency (in particular, self-play) on training efficiency and on the emerged protocol. Here, we use a smaller SHAPES dataset (see B.1), and reduce the vocabulary size and message length ($|\mathcal{V}| = 10$, $L = 3$).

### 6.1 CAN SELF-PLAY REPLACE DIRECTION SUPERVISION?

Here we inspect if self-play supplants direct supervision between agents. We consider the setting in which Alice trains with full data as a speaker, but vary the amount of data she has access to as a listener. We then test Alice's performance as a listener (and vice-versa for Bob as a speaker). Fig. 2 shows the results, with fraction of the full training data that Alice (Bob) sees as a speaker (listener) shown along the x-axis. The self-play models without direct supervision perform well: it appears their protocol transfers across roles with out drifting. This sheds some light on the performance drop between the "one role" and "two role" settings in Section 5.2.2. where the additional experience in the "two role" setting did not help. Fig. 2 shows that additional training in the primary role is unnecessary, so it is not surprising that training on disjoint features (`train-2`) is helpful.

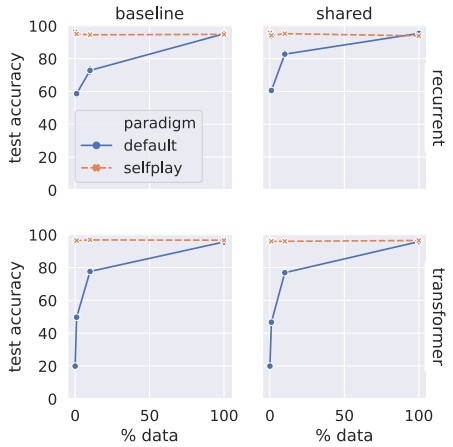

Figure 2: Models trained with self-play perform as well in novel roles as do vanilla models trained with direct supervision in the target role.

### 6.2 DOES SELF-PLAY ENCOURAGE BETTER PROTOCOLS TO EMERGE?

We measure whether self-play leads to better communication protocols in general. First, we measure the agents' speaking and listening capacities separately, using measures proposed by Lowe et al. (2019); Eccles et al. (2019). Specifically, *positive signaling* ($S^+$) measures if the speaker's messages depend on the features of the referent, and *positive listening* ($L^+$) measures if the listener's actions depend on the message[6]. Table 5 shows that self-play improves the agents' communication as measured by accuracy as well as these orthogonal metrics. We also find that the model architectures and

|  | Baseline | | | +Self-play | | | | | |
|---|---|---|---|---|---|---|---|---|---|
|  | % | $L^+$ | $S^+$ | % | $\Delta$ | $L^+$ | $\Delta$ | $S^+$ | $\Delta$ |
| RNN | 95.4 | 4.0 | 6.1 | 96.2 | +0.8 | 6.4 | +1.4 | 6.4 | +0.3 |
| Trans | 95.1 | 2.6 | 5.4 | 96.3 | +1.2 | 3.1 | +0.5 | 5.5 | +0.1 |

Table 5: **Self-play improves the model's communication**: the agents' respective positive listening and positive signaling metrics both improve in the vanilla setting.

self-play impact the agents' lexicons. The recurrent models produces fewer unique messages than the transformer models (on average 110 versus 300), and often neglect to use all the vocabulary. Fig. 3 shows that self-play helps the recurrent model use more of the vocabulary, and leads to both the recurrent and transformer models to develop sparser mappings from symbols onto features.

---

[6]See Appendix B.2 for details.

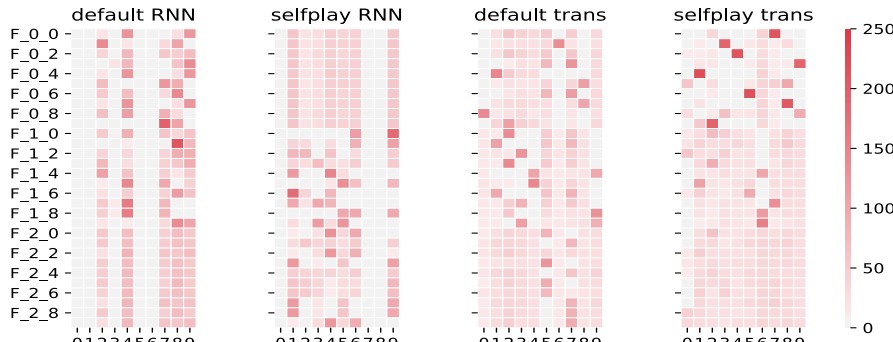

Figure 3: **Emerged lexicons.** This figure show features along the y-axis and symbols along the x-axis. Each index counts the co-occurence of the symbol and the feature. For example, the transformer self-play model appears to map a symbol to each distinct value of the first attribute, ignoring other attributes and word order. See more lexicons over other random seeds in B.3.

## 7 RELATED WORK

Work in emergent communication (Das et al., 2017; Lazaridou et al., 2018) analyzes agents that develop a shared protocol by playing reference games (Lewis, 2008). Kottur et al. (2017) presented results showing that computational models do not learn compositional protocols by default. Instead, the agents tend to develop brittle protocols that have trouble generalizing to novel items. Several approaches have been proposed which could encourage models to learn more generalizable representations of language, including compression (Kirby et al., 2015), efficiency (Gibson et al., 2019), memory constraints (Kottur et al., 2017), pragmatic constraints (Tomlin & Pavlick, 2018), and positive biases to respond to other agents (Jaques et al., 2018; Eccles et al., 2019). Some work, like ours, assumes access to symbolic representations of referents and their attributes, whereas others' are set in pixel-based multi-agent games (Jaques et al., 2018; Eccles et al., 2019; Das et al., 2018) or multi-agent grid-worlds (Sukhbaatar et al., 2016; Lowe et al., 2019).

Our work also relates to a broader body of work on speaker-listener models, specifically pragmatically-informed models in which speakers reason recursively about listeners' beliefs (and vice-versa) (Frank & Goodman, 2012; Goodman & Frank, 2016). Such models have been used in applications such image captioning (Andreas & Klein, 2016; Yu et al., 2017; Monroe & Potts, 2015), and robotics (Vogel & Jurafsky, 2010; Vogel et al., 2013; Fried et al., 2018), as well as in linguistics and psychology in order to explain complex linguistic inferences (Tessler & Goodman, 2016; Monroe et al., 2017). Conceptually, our proposed internal-consistency constraints share something in common with these neural speaker-listener models developed outside of emergent communication. However, again, past work has tended to assume that a speaker's mental model of their listener is not necessarily consistent–in fact, it is often assumed explicitly to be inconsistent (Frank & Goodman, 2012)–with the way the speaker themself would behave as a listener. We note, however, that our proposed model architecture (because it lacks the recursion typical in other pragmatics models) is likely unable to handle the types of higher-level inferences (e.g. implicatures) targeted by the mentioned prior work on computational pragmatics, though this is an interesting avenue to explore.

## 8 CONCLUSION

We propose three methods for encouraging dialog agents to follow "the golden rule": speak/listen to others as you would expect to be spoken/listened to. In the emergent communication setting, we find that the internal-consistency constraints do not systematically improve models' generalization to novel items, but both the self-play objective and shared embeddings significantly improve performance when agents are tested on roles they were not directly trained for. In fact, when trained in one role and tested on another, these internal-consistency constraints allow the agents to perform about as well as if they had been trained in the target role.

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

## A  IMPLEMENTATION DETAILS

We use the deep learning framework Pytorch[7] (v1.2.0) to implement our models (and Python 3.7.3). For reproducibility, all random seeds (random, numpy, torch) are arbitrarily set to 42.

The general architecture, the four modules that comprise each agent are shown in Figure 4.

### A.1  RECURRENT IMPLEMENTATION

The recurrent model decodes and encodes message as follows: to generate a message, the first input is the embedding of a SOS start-of-sentence symbol and the initial hidden state is set as the embedded referent (and the cell memory is all zeroes). From here, at each step, the outputted hidden state ($\in \mathbb{R}^D$) is projected by the transposed word embeddings ($E_{message}^\mathsf{T} \in \mathbb{R}^{D \times |V|}$), and the next word is sampled from this resulting distribution across the vocabulary. Moving forward, the next input is the embedding of the sampled word, and the hidden state and cell memory are set those emitted at the previous step. We produce words until the maximum length is reached.

When encoding a message, a learned embedding is set to the first hidden state, and the input at each time step is the corresponding embedded word. The last hidden state is set as the encoding. In the symmetric variant of this model, the LSTM cell used for encoding and decoding is the same.

---

[7]https://pytorch.org/docs/master/torch.html

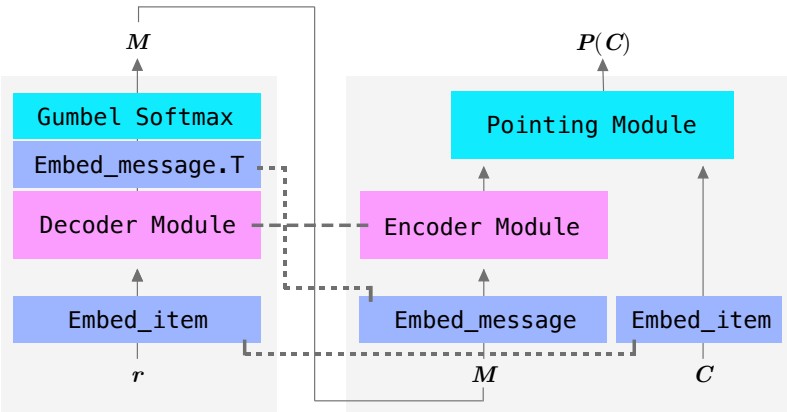

(a) Speaker Modules.  (b) Listener Modules.

Figure 4: **General architecture.** This architecture underlies each model we use; only the implementation of the Decoder and Encoder modules vary between models. In the baseline models no parameters shared within an agent. In shared embedding models, the embeddings (purple) are shared across roles. In symmetric models, the encoder and decoder (pink) are shared across both roles. The blue modules are non-parametric.

## A.2 Transformer Implementation

The transformer model decodes and encodes message as follows: to generate a message describing the referent auto-regressively, all the embeddings of the words produced so far $M$ and the embedding of a NEXT symbol are concatenated together into a matrix $X$ (Eq. 2). Next, the transformer decoder consumes this matrix and the referent embedding and produces a contextualized representation of the input matrix (Eq. 3). The last column vector $\tilde{X}_{:,W}$, which corresponds to the NEXT embedding, is the internal representation of the next word. The next word $m$ is sampled from the projection of this representation with the transposed word embedding. More words are produced in this way until the maximum length message is formed. Producing the $i+1th$ word (so $M$ consists of the first $i$ words), works as follows:

$$X = [E_{message}(M); E_{\text{NEXT}}] \in \mathbb{R}^{(i+1) \times D}, \tag{2}$$

$$\tilde{X} = \texttt{TransformerDecoder}(X, E_{item}(r)) \in \mathbb{R}^{(W+1) \times D}. \tag{3}$$

$$M_{i+1} = \texttt{GumbelSoftmax}(\tilde{X}_{:,i} \times E_{message}^{\mathsf{T}}) \tag{4}$$

To encode incoming messages when listening, all the embeddings of the words in the message plus the embedding of a ITEM symbol are concatenated together into a matrix $X$ (Eq. 5). Then, the transformer encoder is used to produce a contextualized embedding $\tilde{X}$ (Eq. 6). The last column vector $\tilde{X}_{:,W}$, which corresponds to the ITEM embedding, is set as the message encoding. Note, $W$ is the number of words in each message (and the length of $M$).

$$X = [E_{message}(M); E_{\text{ITEM}}] \in \mathbb{R}^{(W+1) \times D}, \tag{5}$$

$$\tilde{X} = \texttt{TransformerEncoder}(X) \in \mathbb{R}^{(W+1) \times D}. \tag{6}$$

$$\hat{r} = \tilde{X}_{:,W} \tag{7}$$

## A.3 Symmetric Agents

For the *Symmetric Recurrent Model*, a LSTM cell is shared between the encoder and decoder. Otherwise, the recurrent model is unchanged. For the *Symmetric Transformer Model*, Transformer Encoders and Transformer Decoders have different structures, so to share parameters between them, we have change either how the transformer agent speaks or how it listens. We opt to replace the Transformer Decoder with Transformer Encoder, and use it to decode messages in-place when speaking.

### A.3.1 SYMMETRIC TRANSFORMER IMPLEMENTATION

The Symmetric Transformer uses the same mechanism for encoding messages when listening as the default transformer model (described directly above). However, it uses a Transformer Encoder when speaking instead of a Transformer Decode. When speaking, to produce the next symbol, the embeddings of the $i$ words produced so far, the referent embedding, and the embedding of a NEXT symbol are concatenated together into a matrix $\boldsymbol{X}$ (Eq. 8). The Transformer Encoder then maps $\boldsymbol{X}$ to a contextualized representation $\tilde{\boldsymbol{X}}$ (Eq. 9). Finally, the column vector in $\boldsymbol{X}$ that corresponds to NEXT, is used to sample the next symbol:

$$\boldsymbol{X} = [E_{item}(\boldsymbol{r}); E_{message}(\boldsymbol{M}); E_{\texttt{ITEM}}], \qquad \boldsymbol{X} \in \mathbb{R}^{(i+2) \times D}, \qquad (8)$$

$$\tilde{\boldsymbol{X}} = \texttt{TransformerDecoder}(\boldsymbol{X}), \qquad \tilde{\boldsymbol{X}} \in \mathbb{R}^{(i+2) \times D}, \qquad (9)$$

$$\boldsymbol{M}_{i+1} = \texttt{GumbelSoftmax}(\tilde{\boldsymbol{X}}_{:,i+1} \times E_{message}^{\intercal}), \qquad \boldsymbol{M}_{i+1} \in \mathcal{V} \subset \{0,1\}^{|V|}. \qquad (10)$$

### A.4 HYPER PARAMETERS

| Search strategy | uniform sampling (25 samples) |
|---|---|
| **Hyperparameter** | **Search Space** |
| optimizer | RMSProp |
| early stopping patience | 100 |
| batch size | 64 |
| number of layers | 1 |
| hidden dimensionality | choice[16, 32, 64, 128] |
| learning rate | choice[0.0001, 0.001, 0.01] |
| scheduler | choice[None, ReduceLROnPlateau, CyclicLR] |

Table 6: Recurrent Model Hyperparameter Search Results.

| Search strategy | uniform sampling (25 samples) |
|---|---|
| **Hyperparameter** | **Search Space** |
| optimizer | RMSProp |
| early stopping patience | 100 |
| batch size | 64 |
| number of layers | 1 |
| hidden dimensionality | choice[16, 32, 64, 128] |
| number of attention heads | choice[1, 2, 8] |
| dropout | choice[0., 0.2, 0.5] |
| learning rate | choice[0.0001, 0.001, 0.01] |
| scheduler | choice[None, ReduceLROnPlateau, CyclicLR] |

Table 7: Transformer Model Hyperparameter Search Space.

We uniformly sampled 25 hyperparameter configurations for each model architecture, experiment, and dataset split. In every case, we fixed the hidden size dimensionality and embedding dimensionality to be the same. We searched over three different learning rate schedulers: (1) None (or no scheduler, (2) ReduceLROnPlateau with a patience of 25, reduction factor of 0.1, and uses validation accuracy as its measure of progress, and (3) CyclicLR rising from 0.00001 to the given learning rate over 500 batches and then declines towards 0.00001 for the rest of training (10,000 batches). This is similar to the Noam Update in Vaswani et al. (2017). To save space, we relegate the hyper-parameter selections in our code bit.ly/internal-consistency-emergent-communication – see /lib/hyperparamters.py.

## A.5 TREE RECONSTRUCTION ERROR

TRE takes two important hyperparameters, an error function and a composition function. We select the same choices as the author, for what amounts to the same task (producing a discrete message) as detailed in the original paper. This method requires structured feature representations, so we assume that the features in each item are entirely right-branching. The composition function is learned, and set the number of update steps to 1000. The original implementation is at `https://github.com/jacobandreas/tre`, and our modification is at `bit.ly/internal-consistency-emergent-communication` in the file `./lib/compositionality.py`.

# B FURTHER ANALYSIS

## B.1 SHAPES SMALL

We simplify the SHAPES dataset in order to be able to empirically compute the positive listening and signaling scores, which requires iterating overall possible messages. In the original version of SHAPES this is impractical as there would be $50^{10}$ possible messages. The details of this smaller version of SHAPES detailed in Table 9.

We fix the settings for the recurrent and transformer models as we found that a majority of models across experiments used the same parameters. See Tables 10, 11. Furthermore, all results in Sec. 6 are averaged over 5 arbitrary random seeds (both trained and tested) (43, 44, 45, 46, 46).

## B.2 POSITIVE SIGNALING AND LISTENING

Positive listening was introduced by Lowe et al. (2019) as Causal Influence of Communication, and the precursor to positive signaling was introduced by Jaques et al. (2018).

We use the definitions Eccles et al. (2019) modified for a one step game for both metrics:

$$\text{Positive Listening } (L^+) \doteq D_{KL}(Pr(a|m) \parallel \overline{Pr_l}(a)),$$
$$\text{Positive Speaking } (S^+) \doteq \mathcal{I}(m, x) = \mathcal{H}(m) - \mathcal{H}(m|x),$$

where $m$ is a given message, $a$ are the actions the agent can take, $x$ is state, $D_{KL}$ is the Kuller-bach divergence, $\mathcal{I}$ is the mutual information, $\mathcal{H}$ is the entropy. We can compute these quantities without sampling messages because the number of messages is tractable.

In our setting, the game is a single step, and the average policy over actions independent of the message converges to the uniform distribution over actions, as the order of referents is randomized. Thus we have: $L^+ = D_{KL}(Pr(a|m) \parallel \mathcal{U})$.

The positive speaking metric is computed over a sampling of the dataset (the distractors are random) as follows as in (Eccles et al., 2019):

$$S^+ = \mathcal{H}(m) - \mathcal{H}(m|x),$$
$$= -\sum_m \overline{\pi}(m) \log \overline{\pi}(m) + E_x[\sum_m \pi(m|x) \log \pi(m|x)],$$

where the summations are over all possible messages, $x$ is the given referent, and $\overline{\pi}(m)$ is empirically likelihood of the message being produced irrespective of $x$, and $\pi(m|x)$ is the likelihood of the given message being produced given the referent $x$. We report empirical averages of $L^+$ and $S^+$ over all items in the dataset, also averaged over 5 arbitrary random seeds.

## B.3 ANALYSIS DETAILS

Figs. 5, 6, 7, 8 show lexicons across random seeds. See Table 8 for additional results in the vanilla setting for SHAPES-small that were elided for space.

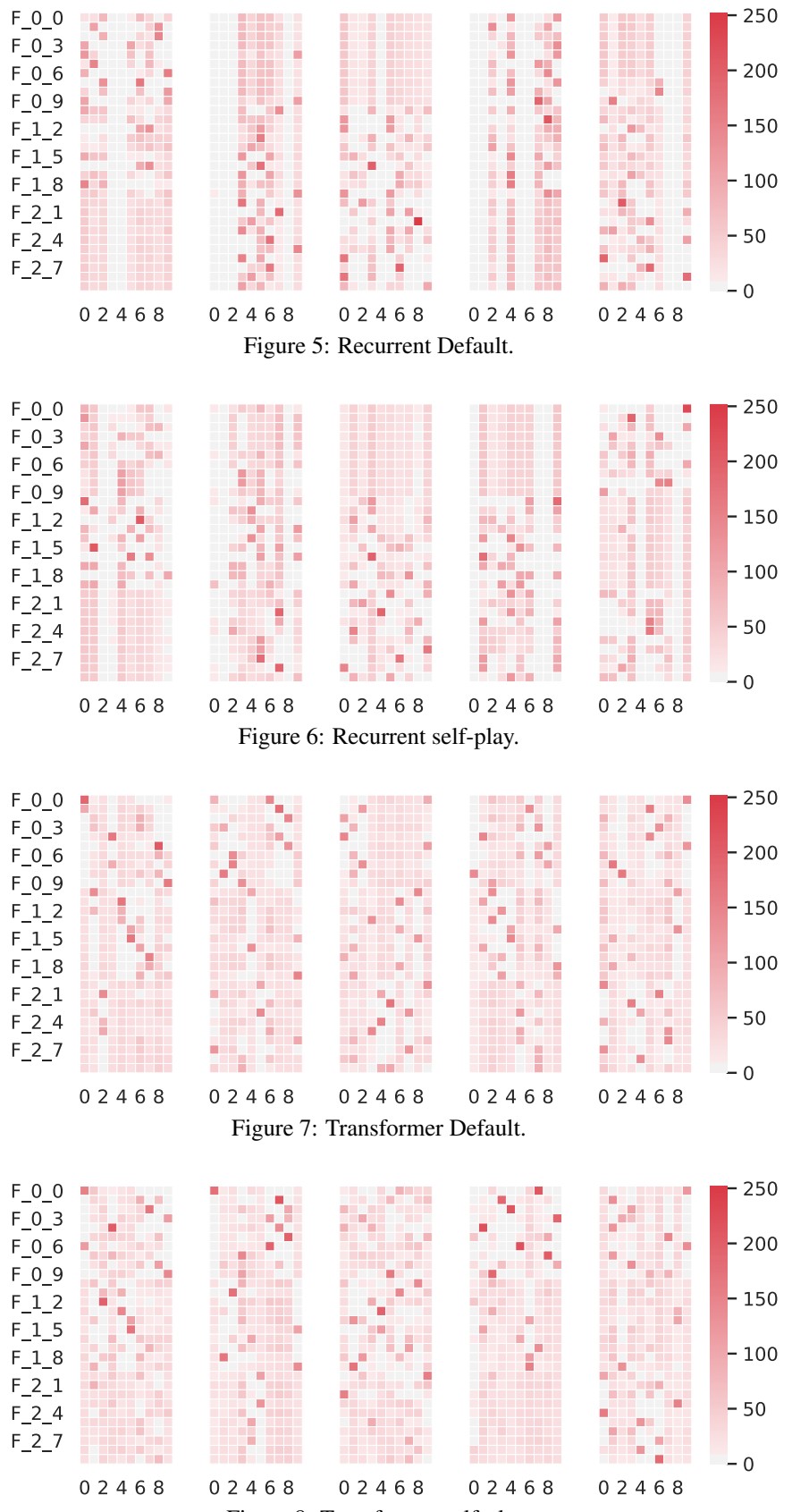

Figure 5: Recurrent Default.

Figure 6: Recurrent self-play.

Figure 7: Transformer Default.

Figure 8: Transformer self-play.

| | Baseline | | | +Self-play | | | +Shared Emb. | | |
|---|---|---|---|---|---|---|---|---|---|
| | % | $L^+$ | $S^+$ | % ,$\Delta$ | $L^+,\Delta$ | $S^+,\Delta$ | %, $\Delta$ | $L^+,\Delta$ | $S^+,\Delta$ |
| RNN | 95.4 | 4.0 | 6.1 | 96.2, +0.8 | 6.4, +1.4 | 6.4, +0.3 | 95.6, +0.2 | 5.2, +1.2 | 6.3, +0.2 |
| Trans | 95.1 | 2.6 | 5.4 | 96.3, +1.2 | 3.1, +0.5 | 5.5, +0.1 | 96.2, +1.1 | 3.4 + 0.8 | 5.6, +0.2 |

Table 8: Deltas are compared to the baseline value for each row. Here we report the results for shared embeddings as well as self-play.

| Name | Features ($n_f$) | Values ($n_v$) | Context Size ($k$) | Train | Val | Test |
|---|---|---|---|---|---|---|
| SHAPES SMALL | 3 | 10 | 5 | 800 | 100 | 100 |

Table 9: Dataset statistics.

| Hyperparameter | Search Space |
|---|---|
| optimizer | RMSProp |
| early stopping patience | 100 |
| batch size | 64 |
| number of layers | 1 |
| hidden dimensionality | 128 |
| learning rate | 0.001 |
| scheduler | None |

Table 10: Best Recurrent Model Hyperparameter Search Settings.

| Hyperparameter | Search Space |
|---|---|
| optimizer | RMSProp |
| early stopping patience | 100 |
| batch size | 64 |
| number of layers | 1 |
| hidden dimensionality | 128 |
| number of attention heads | 1 |
| dropout | 0. |
| learning rate | 0.0001 |
| scheduler | ReduceLROnPlateau(25 epochs) |

Table 11: Best Transformer Model Hyperparameter Settings.

