# OpenReview forum: "INTERNAL-CONSISTENCY CONSTRAINTS FOR EMERGENT COMMUNICATION"
_ICLR.cc/2020/Conference — Reject_

### Official Review · AnonReviewer3 · 2019-10-23
**Official Blind Review #3**

**Rating:** 6

**Review:**

The paper analyzes if enforcing internal-consistency for speaker-listener setup can (i) improve the ability of the agents to refer to unseen referents (ii) generalize for different communicative roles. The paper evaluates a transformer and arecurrent model modified with various sharing strategies on a single-turn reference game. Finally, the paper claims that results with self-play suggest that internal consistency doesn’t help (i) but improves cross-role performance for (ii).

As a reader, the paper doesn’t provide me a concrete finding which can help in designing future multi-agent systems. Most of the results for the experiments (except self-play) don’t have a uniform signal across the board to deduce whether the internal-consistency works in all of the cases. Most of the speaker-listener scenarios emerge in dialog based settings which are multi-turn and require agents to act both as speaker and listener. Though paper advocates through some of its results that self-play is helpful in generalization across roles via internal-consistency, without multi-step experiments, qualitative and quantitative analysis of what is happening and why there is so much variation, the paper is weak in its current form. Therefore, I recommend weak reject as my rating. Below I discuss some of the concerns in detail:

Without multi-step evaluation, it is hard to gauge the extent to which self-play for internal consistency help in generalization of the roles. For e.g., task from Das et. al. (2017) [1] provides a clear signal on how well the agents are able to communicate through dialog evaluation. So in 5.2.1, the setup which requires training in both roles can provide better signal overall if it was trained to do multi-step conversation.

Paper is missing any kind of quantitative or qualitative analysis. What are the differences between the embeddings of the agent that learned via self-play and the one which learned directly. It also be interesting to see how the shared embeddings and symmetric encoding and decoding affect these embedding and might help explain the drop and randomness. In Table 4., the results on symmetric encoding suggest that the claim of generalization through internal consistency might not hold everywhere. For Shared Embedding results, on RNN shapes, it is surprising that training in one role improves performance through internal consistency while in both roles it drops. These require further analysis to solidify the claim. Given the flaky results, to boost the claim, have authors tried other settings to test internal-consistency like Predator-Prey?

Things that didn’t affect the score:

Related work section is missing the relevant discussion on continuous communication work and discussion on why internal consistency wasn’t tested on those settings as well. (See Singh et.al [2]., Sukhbaatar et.al. [3], Das et.al. [4] etc)

The number of pages are above eight, you should reduce the redundancy between table descriptions and text and maybe squeeze Section 2, decrease setup explanation.

The setup for training and test sets explained at the end of page 7 isn’t very clear to me and needs to be rephrased.

[1] Das, Abhishek, Satwik Kottur, José MF Moura, Stefan Lee, and Dhruv Batra. "Learning cooperative visual dialog agents with deep reinforcement learning." In Proceedings of the IEEE International Conference on Computer Vision, pp. 2951-2960. 2017.
[2] Sukhbaatar, Sainbayar, and Rob Fergus. "Learning multiagent communication with backpropagation." In Advances in Neural Information Processing Systems, pp. 2244-2252. 2016.
[3] Singh, Amanpreet, Tushar Jain, and Sainbayar Sukhbaatar. "Learning when to communicate at scale in multiagent cooperative and competitive tasks." arXiv preprint arXiv:1812.09755 (2018).
[4] Das, Abhishek, Théophile Gervet, Joshua Romoff, Dhruv Batra, Devi Parikh, Michael Rabbat, and Joelle Pineau. "Tarmac: Targeted multi-agent communication." arXiv preprint arXiv:1810.11187 (2018).

=========
Post-rebuttal Comments
=========
Thanks for updating the manuscript to resolve my and R2's concerns. The new analysis section does provide good insights into what exactly is happening.

When I was talking about actionable insights, I was talking about both negative and positive insights. Currently, the only take-away is that self-play helps in generalizing to listener roles as well. For the other negative insight that internal consistency doesn't help with generalization, as R2 suggested, it is unclear why that would be case in the first place (I read the pscyhology arguments, but I am not still not convinced). I still believe that without multi-step communication, the work is as useful as it can be in current form. In real world, no meaningful conversation is usually one step.

For Predator-Prey setup, I was talking about OpenAI https://github.com/openai/multiagent-particle-envs in which multiple tasks can be setup. For e.g. if prey thinks of what action predator might take, does internal consistency help prey to perform better?

I think most of what you got is correct for multi-step, see second para for more details in this response.

Thanks for bringing the manuscript under 8 pages. [3] is still missing from references.

Final comments: I would like to see multi-step experiments due to the reasons I explained above. The scheme of internal-consistency should be applicable beyond conversation to Predator-Prey setups also, thus, I feel experiments are not enough (only on 1 setting) to claim generalization of the hypothesis. Beyond these comments, I feel this is still a step in right direction and I would like to update my rating to weak accept while hoping that authors try to address these issues in camera-ready version if paper gets accepted.





**Experience Assessment:**

I have published one or two papers in this area.

**Review Assessment: Checking Correctness Of Derivations And Theory:**

N/A

**Review Assessment: Checking Correctness Of Experiments:**

I assessed the sensibility of the experiments.

**Review Assessment: Thoroughness In Paper Reading:**

I read the paper thoroughly.

---

> ### Author Response · Authors · 2019-11-15
> **Response to Reviewer 3**
>
> Please see our general reply to all reviewers above, as we reply to many of your points there. We also added several new analysis to the paper in reply to your comments; these can be found in Section 6 of the updated draft. In addition, we provide several direct replies to your comments below.
>
> * Why no multi-step communication?
>
> We understand your concern about not using multi-step communication as follows: the agents may behave similarly to agents that self-play if they had to act over multiple turns and correspondingly — as part of the game itself — both speak and listen.
>
> This is a great idea and may extend the range of the results. We’d love to pursue something along these lines in future work. As a starting point, we chose to focus on tasks like reference games, with an applied setting of instruction-following robots. In these contexts, there are often no follow ups to “get me my coffee cup” or “I’d like to buy that red sweatshirt” other than proffering the predicted item within the context. Given that real interactions with humans are expensive (much more so than self-play), it would be beneficial if learning agents could leverage pre-existing data and self-play in order generalize to untrained roles. We did not flesh out this viewpoint because this work does not fully extend into this setting, but your concerns suggest that we should have clarified this.
>
> * What are the differences between the embeddings of the agent that learned via self-play and the one which learned directly? (e.g more qualitative and quantitative analysis!)
>
> We did not heavily inspect the embeddings of the agents, however we did add a section (now Section 6) of additional qualitative and quantitative analysis that inspects the efficacy of selplay, the agents’ communication, and finally, shows the lexicon the agent’s produced.
>
> * For Shared Embedding results, on RNN shapes, it is surprising that training in one role improves performance through internal consistency while in both roles it drops. Why?
>
> It appears that the additional signal does not provide useful information to the agents: perhaps the resultant protocols do not utilize the lexicon developed by the agents in the original role. This would be the case if there were a one-to-one mapping between features and symbols (or something like this). In the new analysis section (Section 6), we inspect this further.
>
> * Follow up
> - Can you elaborate the mentioned predator-prey setting (or point us in the direction you were thinking of)?
> - Is our interpretation of your concern with multi-step games correct?
> - Thank you for highlighting additional related work and for stressing the 8-pages. We worked to amend the report to address both of these concerns.

---

### Official Review · AnonReviewer2 · 2019-10-23
**Official Blind Review #2**

**Rating:** 3

**Review:**

This paper investigates the question of internal consistency in emergent communication. In other words, the paper aims to answer the question ‘how is emergent communication improved if we enforce the constraint that an agent must speak in the same way that it listens?’ The paper explores three methods of enforcing internal consistency: self-play, shared embeddings, and symmetric encoding/decoding. They find that, while internal consistency does not help generalization to unseen objects, it does allow agents to generalize over conversational roles (i.e. to perform well as a speaker despite only being trained as a listener).

I have been eagerly anticipating a paper on this topic – it seems silly that the listening and speaking modules in traditional emergent communication research are completely disjoint. I think coming up with methods/ architectures that combine these two capabilities is an important research direction. I also think paper is very well-written and structured, and it reads very well.

However, I have several concerns with the paper. First, half of the results center around the hypothesis ‘internal consistency helps agents generalize to unseen items’. While ultimately this hypothesis is disproven, it’s unclear as to why this might be expected in the first place. The only justification of this hypothesis I could find in the paper is the sentence “It is conceivable that <internal consistency> might improve performance even though each <agent> remains in a fixed role.” In my view, the fact that this hypothesis is ‘conceivable’ is a rather weak argument for it to be such a prominent part of the paper. Thus, I don’t think this half of the results add much to the overall paper.

I also have mixed feelings about the use of ‘self-play’ to enforce internal consistency, and how it relates to the core result of the paper: “the proposed constraints enable models to generalize learned language representations across communicative roles, even in the case of where the agent receives no direct training in the target (test) role”. In short, I think the phrase “no direct training” is misleading, as the self-play itself is almost a form of direct training, and thus the result isn’t very surprising.

 More specifically, each agent ‘Alice and Bob’ are composed of two modules: a speaking module and a listening module. During normal training (say, in the shape environment), Alice speaks and Bob listens (achieving a reward if Bob selects the right shape, which is backpropagated to Alice), and thus the Alice’s speaker module and Bob’s listener module are updated. Alice’s listening capabilities will be equivalent to a random agent, as her listening module is still randomly initialized. Now, during self-play, Alice’s speaker module is trained with her listener module (I believe in the same way as it is trained with Bob’s listener module) to achieve high reward. This listener module is then tested against Bob’s speaker module (which is also trained via self-play). To me, this process isn’t the same as the paper’s narrative of ‘we only train Alice to be a speaker, and she learns to listen!’ This is especially true since, without parameter tying, the choice of saying that listener module ‘belongs to Alice’ is arbitrary (since it’s completely separate). An equivalent way of framing this result would be “language learning is transitive: if we train agent A to speak to agent B who listens, then train agent C to listen to agent A and agent D to speak to agent B, then agents D can perform well with agent C”. With this framing, the result is much less surprising (in fact, it would be surprising if this weren’t true).


Finally, the three methods of enforcing internal consistency are not tested independently --- shared embeddings are only tested on top of self-play, and symmetric encoding/decoding is only tested on top of the other two. While this does make the paper more concise, I suspect another reason for this is that the self-play is the core driver of performance, and without it the other two methods don’t do much. I’d like this to be explained more explicitly in the paper.

Overall, I really like the problem the paper is tackling, however I have some issues with the framing of the paper in relation to the self-play constraint, and subsequently with the importance of the results. Thus, I do not recommend acceptance in the paper’s current form.


Questions:
-	“We set shared embedding agents to always use the self-play objective, because otherwise its equivalent to the baseline agent” -> it’s not clear to me why this is the case. Can this be elaborate on?
-	Shouldn’t the final row of Table 4 read ‘Trans, shapes’?

Small fixes:
-	This assumption is a reasonable -> is reasonable
-	Section 5.1.2: “We observe a no clear trend associated with the shared embedding module (sometimes it helps, sometimes it hurts)” -> I don’t see any results on shared embeddings in Table 2?


**Experience Assessment:**

I have published in this field for several years.

**Review Assessment: Checking Correctness Of Derivations And Theory:**

N/A

**Review Assessment: Checking Correctness Of Experiments:**

I carefully checked the experiments.

**Review Assessment: Thoroughness In Paper Reading:**

I read the paper at least twice and used my best judgement in assessing the paper.

---

> ### Author Response · Authors · 2019-11-15
> **Response to Reviewer 2**
>
> Please see our general reply to all reviewers above, as we reply to many of your points there. In addition, we provide several direct replies to your comments in this post.
>
> * What is the motivation for the hypothesis that internal consistency will improve generalization to unseen items?
>
> You raised concerns that our first hypothesis (internal consistency can improve generalization to unseen items) is not well motivated. We disagree. Our initial motivation for this line of work was based largely on evidence from developmental psychology which suggests that pragmatic reasoning may encourage children to develop “one-to-one” mappings between words and referents (see e.g. [1]). These types of one-to-one lexicons are what we want to see agents learn in emergent settings, and is often what is actually meant when people use words like “compositional” or “grounded” in other similar EC work. We deliberately chose *not* to lean on the psychological motivation in this paper because we believe that, presently, the ML field is a bit too quick to try to connect the models we build to arguments about “how the brain works”. We are happy to draw inspiration from evidence in psychology and believe others absolutely should too. But we prefer to minimize the risk that papers are misread as evidence for/against a hypothesis about human language acquisition unless the experiment is in fact designed precisely to test psychological theories. Thus, we chose to take a more conservative approach to our framing by letting the ML hypotheses speak for themselves, rather than frame them as somehow entwined with hypotheses about human language acquisition. We still believe that this was the appropriate choice, but hope you can see that the hypothesis was in fact motivated, not pulled out of thin air. We rephrased the first paragraph of Section 5.1.3 to provide a bit more motivation, though still without pulling in psych arguments. We hope you are happier with this wording.
>
> [1] Diesendruck and Markson. Children's Avoidance of Lexical Overlap: A Pragmatic Account. Developmental Psychology 2001. Vol 37. No. 5, 630-641.
>
>
> * Why are shared embeddings and symmetric (de)encoders not tested in isolation?
>
> We now think our presentation, and perhaps terminology, are not sufficiently clear.
>
> Both shared embedding and symmetric decoders/encoders are instances of parameter-tying. Note that parameter-tying only matters if the agents use these tied parameters. So, if the agents only ever train/test in a signal role, then the weights tied between roles have no impact. Therefore “we set shared embedding agents to always use the self-play objective” because otherwise the “constrained” model would be no different from the baseline model.
>
> We realize that this is not the case in a wider range of experimental setups (that go beyond the scope of our paper): for example, if the agents were set to both speak and listen to each other, then indeed there would be a distinction. This setup somewhat occurs in our experiment on role generalization where Alice listens to Bob on “shapes 1”, Alice speaks to Bob on “shapes 2”, and then is tested on speaking to Bob on “shapes 1”. However, as we noted without self-play all methods achieve poor performance in role generalization, and we now highlight this in the report.
>
> * Are we going beyond determining if language learning is transitive?
>
> In light of the previous question, it is not the same because we evaluate if parameter-tying impacts the result. Thus, with this framing in mind, we are investigating if there is any interplay between self-play and parameter-tying that goes beyond transitive language learning. It appears, especially for role generalization, internal consistency constraints (self-play + parameter-tying) go beyond transitive language learning in many cases. In any case, it’s not evident (without our evaluation) that the protocols would not drift between roles/agents.
>
> * Further updates
> - We tried to highlight that “direct supervision” refers to interaction between the agents evaluated at test time. We found it apt because the other interaction (self-play) is auxiliary, and from the perspective of the final task it is unsupervised.
> - Yes, the final row of table 4 should be labeled “Trans, shapes” — thank you, we fixed this in our revised report.

---

### Official Review · AnonReviewer1 · 2019-10-25
**Official Blind Review #1**

**Rating:** 3

**Review:**

In an emergent communication setting, the paper proposes three methods for encouraging internal-consistency for dialog agents --- an agent speaks/listens to others as it would expect to be spoken/listened to. The paper concludes that the internal-consistency constraints do not improve models’ generalization to novel items, but improve performance when agents are tested on roles they were not trained for.

The experiments support the above conclusions on the small datasets used in the experiments. The paper is in general well structured and is clear and easy to follow.

The contributions of the paper are rather incremental. The main methods are described in section 3.3, which employ a self-play objective,  use shared embedding, and include sysmetric encoding and decoding. The extension is new but does not contain enough content for the conference. I do not think this paper presents enough contributions.


**Experience Assessment:**

I have read many papers in this area.

**Review Assessment: Checking Correctness Of Derivations And Theory:**

I assessed the sensibility of the derivations and theory.

**Review Assessment: Checking Correctness Of Experiments:**

I assessed the sensibility of the experiments.

**Review Assessment: Thoroughness In Paper Reading:**

I read the paper at least twice and used my best judgement in assessing the paper.

---

### Author Response · Authors · 2019-11-15
**Responses to AC+all reviewers**

Thank you to R2 and R3 for the very thoughtful feedback. As we see it, there are two overarching issues raised about the work: 1) how we framed the work in terms of both motivation and the hypotheses, and 2) that the results are not sufficiently impactful. We believe that (2) largely stems from framing, and thus respond to both criticisms jointly, below. We provide specific responses to Reviewers 2 and 3 in their own comments.

- Just because the paper presents negative results does not mean it is not useful: R2 emphasized the importance of the topic addressed by our work, i.e. that treating speaking and listening as disjoint functions (as in prior work) is unsatisfying and is fundamentally wrong. R3 complains that the paper doesn’t provide actionable insights for building multi-agent systems. These comments together, we believe, emphasize why the paper is scientifically impactful. We are the first (as far as we know) to attempt to operationalize the symmetric relationship between the speaking and listening tasks. We chose several natural architectural modifications for doing so. That those modifications did not yield uniformly, overwhelmingly positive results is, we agree, unfortunate. We would have preferred clear performance wins across the board as well! :) But as is: we tested several interesting, relevant, motivated hypotheses using sound experimental design. We don’t claim that this paper closes the book on the topic, rather it opens questions for future work, and that is precisely what makes the work interesting to the community.

- Distinction between direct supervision and “self play” is important: Our results show that using internal consistency constraints (manifested as self-play + parameter-tying), helps agents transfer “knowledge” across roles. The reviewers were underwhelmed by this result. This was our fault for not framing as clearly as we could have. The distinction between “direct supervision” (feedback from another agent) and self play is very consequential in practice, as we expect to demonstrate experimentally in future work. Specifically, consider the case when the “other agent” is a human. In this case, the AI agent may receive very limited interaction and thus minimal “direct supervision” in one role. E.g. it is very likely that humans are willing to train robots by speaking to them, but not by listening to them (we don’t want to follow noisy commands given by a partially-trained robot). In such a setting, the ability to use self play to substitute is, we believe, non-obvious and very exciting for the development of systems going forward. We note that a very similar observation was made in another submission to this conference this year (https://openreview.net/pdf?id=rJxGLlBtwH), albeit using a different setup. The fact that we are not the only lab that finds this topic interesting further supports that the interplay between these training settings is interesting and the results are non-obvious. As such, negative results will provide useful information to the community going forward.

- Updated draft: We have updated the paper with new supporting analyses (Section 6) and some refinements to the narrative, as requested by R2 and R3. We provide specific pointers in individualized responses to each reviewer. In particular, the new analysis 1) reiterate the point that self-play serves as a substitute for direct supervision, and 2) corroborate the claim that self play improves communication overall, using several measures other than just accuracy on the test set.

---

### Decision · Program_Chairs · 2019-12-19

**Decision:**

Reject

**Comment:**

This work examines how internal consistency objectives can help emergent communication, namely through possibly improving ability to refer to unseen referents and to generalize across communicative roles. Experimental results support the second hypothesis but not the first.
Reviewers agree that this is an exciting object of study, but had reservations about the rationale for the first hypothesis (which was ultimately disproven), and for how the second hypothesis was investigated (lack of ablations to tease apart which part was most responsible for improvement, unsatisfactory framing). These concerns were not fully addressed by the response.
While the paper is very promising and the direction quite interesting, this cannot in its current form be recommended for acceptance. We encourage authors to carefully examine reviewers' suggestions to improve their work for submission to another venue.